# Personalized MRI-based characterization of subcortical anomalies in Ataxia-Telangiectasia using deep-learning

Catalina Saini[1]☯, Cristian Salazar-Vilches[2,3]☯, Caroline C.V. Blanchard[4],
William P. Whitehouse[5,6], Denis Parra[2,7], Rob A. Dineen[4,8,9], Stefan Pszczolkowski[4,8,9]*

**1** College, Pontificia Universidad Católica de Chile, Av. Vicuña Mackenna, Macul, Santiago, Chile,
**2** Instituto Milenio iHealth, Av. Vicuña Mackenna, Macul, Santiago, Chile, **3** Faculty of Medicine,
Pontificia Universidad Católica de Chile, Av. Libertador Bernardo O'Higgins, Santiago, Santiago, Chile,
**4** Radiological Sciences, School of Medicine, University of Nottingham, Queen's Medical Centre,
Nottingham, United Kingdom, **5** Paediatric Neurology, Nottingham Children's Hospital, Nottingham
University Hospitals NHS Trust, Queen's Medical Centre, Nottingham, United Kingdom, **6** School of
Medicine, University of Nottingham, Queen's Medical Centre, Nottingham, United Kingdom, **7** Department
of Computer Science, Pontificia Universidad Católica de Chile, Av. Vicuña Mackenna, Macul, Santiago,
Chile, **8** NIHR Nottingham Biomedical Research Centre, Queen's Medical Centre, Nottingham, United
Kingdom, **9** Sir Peter Mansfield Imaging Centre, University of Nottingham, University Park, Nottingham,
United Kingdom

☯ CS and CSV contributed equally to this work.
* stefan.pszczolkowskiparraguez@nottingham.ac.uk

journal.pone.0328828

Hospital, UNITED STATES OF AMERICA

**Peer Review History:** PLOS recognizes the
benefits of transparency in the peer review
process; therefore, we enable the publication
of all of the content of peer review and
author responses alongside final, published
articles. The editorial history of this article is
available here: https://doi.org/10.1371/journal.
pone.0328828

## Abstract

### Background

Cerebellar atrophy is a known feature of ataxia-telangiectasia (A-T). However,
basal ganglia dysfunction contributing to extrapyramidal movement disorders in A-T
remains understudied.

### Objectives

To characterize basal ganglia abnormalities in A-T using a normative self-supervised deep
autoencoder trained on MRI-based diffusion and perfusion features from healthy children.

### Methods

Mean values of apparent diffusion coefficient and cerebral blood flow perfusion maps
were extracted from seven regions-of-interest: caudate, hippocampus, pallidum,
putamen, thalamus, cerebellar gray matter and cerebellar white matter. A normative
deep autoencoder that reconstructs these features was trained on healthy subjects.
Reconstruction errors for healthy and A-T participants were computed. We used
Shapley Additive Explanations (SHAP) to identify the most influential features contrib-
uting to the features' reconstruction predictions. Correlations between reconstruction
errors and clinical scores in A-T patients were evaluated.

**Data availability statement:** https://github.com/stefanpsz/A-T_Autoencoder.

**Funding:** The author(s) received no specific funding for this work.

**Competing interests:** The authors have declared that no competing interests exist.

## Results

Features were correctly reconstructed in controls but not A-T participants, who showed significantly higher reconstruction errors. Hippocampus, caudate and putamen diffusion, and caudate and putamen perfusion were overestimated, and cerebellar diffusion and pallidum perfusion underestimated, in participants with A-T. SHAP scores revealed that caudate, putamen, and hippocampus perfusion had the greatest influence on the reconstruction of perfusion features. In contrast, cerebellar diffusion and caudate perfusion had the greatest influence on the reconstruction of diffusion features. Exploratory analysis showed that extrapyramidal movement sub-scores from A-T participants correlated with perfusion and diffusion reconstruction errors from cerebellar and subcortical structures.

## Conclusion

Our findings suggest that pallidum, caudate, and cerebellar gray matter are potential targets for novel treatment approaches for A-T. The approach enables identification of subtle tissue anomalies at an individual level, allowing tailored approaches.

---

## 1. Introduction

Ataxia-telangiectasia (A-T) is a rare autosomal recessive disorder caused by mutations in the *ATM* gene, which codes for the ATM protein kinase [1,2]. Absent or diminished ATM function leads to impaired cellular responses to oxidative damage and DNA repair. Features of A-T include progressive neurological dysfunction, cancer predisposition, immunodeficiency, respiratory problems, cutaneous telangiectasia, and reduced life expectancy [3]. Progressive cerebellar ataxia is a central feature, as well as extrapyramidal movement disorders, including bradykinesia, dystonia, and hypermotor [4–7].

Previous neuroimaging studies focus on cerebellar atrophy in A-T [8]. However, the frequency of extrapyramidal movement disorders also implies dysfunction in basal ganglia motor circuits. Perfusion and diffusion magnetic resonance imaging (MRI) provide non-invasive, non-ionizing methods for quantifying tissue changes in the cerebellum and basal ganglia in people with A-T. To date, the perfusion and diffusion characteristics of the basal ganglia and their relationship to movement abnormalities in A-T remain unexplored.

Dineen et al. showed increased cerebellar apparent diffusion coefficients (ADC) for A-T, reflecting loss of ultrastructural barriers to diffusion secondary to neurodegeneration [9]. Previous animal models of A-T have shown that nigrostriatal degeneration is detectable in ATM-deficient mice [10,11]. We postulated that ADC may provide an index of ultrastructural changes in the basal ganglia reflecting degeneration in A-T. We also postulated that perfusion changes may occur in the basal ganglia following prior work showing altered basal ganglia metabolism [12], on the basis that blood perfusion and metabolic activity are coupled.

Artificial intelligence (AI) shows potential for classifying and characterizing pathologies [13,14]. In this work, we use an autoencoder to identify anomalies in basal ganglia diffusion and perfusion features in a personalized way. An autoencoder is a neural network for unsupervised machine learning and deep learning [15]. The encoder architecture reduces input data dimensionality by creating a compact representation, encoding meaningful patterns from complex data into a lower-dimensional form (the latent layer). The decoder then reconstructs the original input from the encoded representation as closely as possible. Autoencoders are trained to minimize a loss function based on reconstruction errors of each feature when encoding and decoding input data. A common use of autoencoders is for detecting anomalies relative to a normative model. Normative models are created by learning typical patterns of representation within control training data [16], encoding only essential data features. When an autoencoder is trained on control data, it will result in higher reconstruction errors when tested on non-controls because it cannot correctly reconstruct out-of-distribution data. This discrepancy in reconstruction errors serves as a basis for identifying anomalies [17]. Moreover, since the input to an autoencoder corresponds to feature data from a single subject, it allows participants to be analyzed genuinely individually, steering away from classical group-wise statistical analyses [18].

In this retrospective exploratory study of a prospectively collected imaging dataset, we (**a**) employ self-supervised anomaly detection to identify children with A-T based on a normative model of healthy controls, (**b**) compare differences in reconstruction errors of subcortical and cerebellum perfusion and diffusion data of controls and participants with A-T, (**c**) use an explainability framework to understand features driving anomalies in participants with A-T, and (**d**) perform a correlation analysis of reconstruction errors with hypermotor symptoms, bradykinesia, dystonia, and ataxia.

## 2. Materials and methods

### 2.1. Participants and recruitment

Twenty-five A-T participants were recruited to the Childhood A-T Neuroimaging Assessment Project (CATNAP) phases 1 and 2 from the National Paediatric A-T Clinic at the Nottingham Children's Hospital. Inclusion criteria were age 3–18 years at their first visit, with a confirmed diagnosis of A-T. Exclusion criteria were contraindication to MRI, current or previous cancer or cancer treatment, or other (non-A-T) neurological or neurosurgical conditions. CATNAP participants underwent neurological assessment using the A-T NEST scale [19], which includes scores for ataxia, hypermotor, bradykinesia, and dystonia. A lower score indicates a more severe neurological disorder.

Twenty-seven healthy control volunteers were also recruited to CATNAP. Inclusion criteria were 3–18 years of age at the time of their first visit. Exclusion criteria were contraindications to MRI, neurological or neurosurgical conditions, or significant medical conditions. Additional data from 120 healthy control subjects were obtained from the publicly available Calgary Preschool MRI dataset [20]. Inclusion criteria were 2–8 years of age at the time of recruitment (**Fig 1**).

Research Ethics Committee approval for CATNAP was granted by the UK National Health Service Research Ethics Service (phase 1: ref. 14/EM/1175; phase 2: ref. 18/SW/0078). Written informed consent was obtained from participants aged 16–18 years, and from parents/guardians of participants aged under 16 years, witnessed by a senior member of the research team and archived in the study site file.

The Calgary Preschool MRI dataset is an open-access dataset that can be accessed from the Open Science Framework at https://osf.io/axz5r/. Therefore, no ethics approval was needed to use this dataset in this research.

For the purpose of this research, all data was accessed and feature spreadsheets generated on the 22nd of April 2024. Some authors (C.C.V.B., W.P.W., R.A.D., S.P.), had access to information that could identify individual participants during and after data collection.

### 2.2. MRI protocol

Scanning was performed on 3T Discovery MR750 (GE Healthcare, Milwaukee, WI) with a 32-channel head coil without sedation for both CATNAP and Calgary datasets. In CATNAP, in addition to standard pediatric MRI preparation, younger

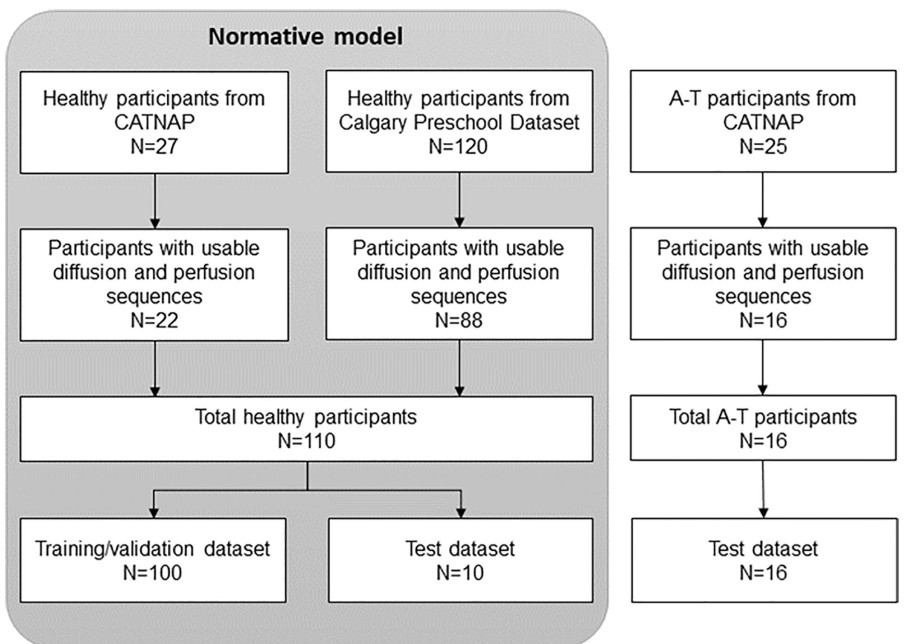

**Fig 1. Flow diagram of participant enrollment.** (A-T = Ataxia Telangiectasia, CATNAP = Childhood A-T Neuroimaging Assessment Project).

participants were shown an animation to prepare them for MRI [21] and were scanned watching a movie via an MRI compatible audiovisual system. For the Calgary dataset, children were scanned while watching a movie or sleeping [20]. The acquired sequences include structural T1-weighted (T1w) imaging, pseudo-continuous arterial spin labeling (pCASL), diffusion-weighted imaging (DWI), and/or diffusion tensor imaging (DTI). Details of the acquisition protocol for CATNAP are provided in the supplementary materials (**Appendix A1 in** S1 File). For acquisition details on the Calgary dataset, refer to Reynolds et al. [20].

## 2.3. Image analysis

Structural T1w images were bias-corrected using N4 [22], brain extracted with ROBEX [23], and segmented into 138 labels of interest using MALP-EM [24]. Seven regions of interest (ROIs) were derived from the segmentations: pallidum, caudate, putamen, thalamus, hippocampus, cerebellar white matter (CWM), and cerebellar gray matter (CGM) (composed of cerebellar cortex and vermis). ADC maps ($10^{-6}$ mm$^2$/s) were computed from the DWI data or from three orthogonal directions for DTI data-only subjects [9]. Raw cerebral blood flow (CBF) maps (mL/100 g/min) were computed from pCASL pairs, and partial volume (PV) effects were corrected using the mLTS method [25]. The GM and WM CBF maps were smoothed with a Gaussian kernel (4 mm FWHM). Due to spatial distortions, ADC maps were transformed into structural space by non-linear registration. CBF maps were transformed to structural space using the inverse of the boundary-based rigid transformation between the raw CBF map and the GM PV estimation in structural space resulting from mLTS correction [26]. The final CBF map corresponds to the intensities of the transformed smoothed GM CBF map of voxels labeled as GM by the mLTS method and analogously for WM voxels.

Using the defined ROIs, we computed seven diffusion features from the ADC maps (caudate, hippocampus, pallidum, putamen, thalamus, CGM, and CWM mean) and six perfusion features from the CBF maps (caudate, hippocampus, pallidum, putamen, thalamus, and CGM mean), and seven covariates (age, sex, cerebral volume (mL), cerebellar volume (mL), age$^2$, age × sex, and age$^2$ × sex). We normalized diffusion and perfusion features by dividing their values by 3,000

and 100, respectively, considering their quantitative scales. Covariate variables were rescaled to a $[0 - 1]$ range using min-max normalization.

## 2.4. Development of an unsupervised deep learning autoencoder

An autoencoder was implemented using PyTorch [27]. The architecture comprises seven asymmetrical, fully connected layers. The input layer comprises 20 features (six perfusion features, seven diffusion features, and seven covariates). The hidden layers have 15, 10, 4, 8 and 10 neurons. The output layer has 13 features, so the model reconstructs the imaging features only (**Fig 2**). ReLU activation was used between layers to promote sparse activation.

We minimized the median Mean Squared Error (MSE) loss function to train and evaluate the quality of the reconstructions. The MSE calculates the average squared reconstruction errors across features. These errors correspond to the difference between the original and reconstructed feature values, thereby amplifying larger deviations to capture discrepancies sensitively. This ensures that even subtle differences are reflected. MSE for each subject is given by:

$$MSE = \frac{1}{n} \sum_{i=1}^{n} (y_i - \hat{y}_i)^2 ,$$

(1)

where $y_i$ and $\hat{y}_i$ are the original and reconstructed values for the $i$-th feature, and $n$ is the number of features.

Hyperparameters for the autoencoder were optimized using grid search and cross-validation **(Appendix A2 in S1 File)**. We trained the normative model using only healthy controls (100 for training and validation, and 10 for testing). We tested

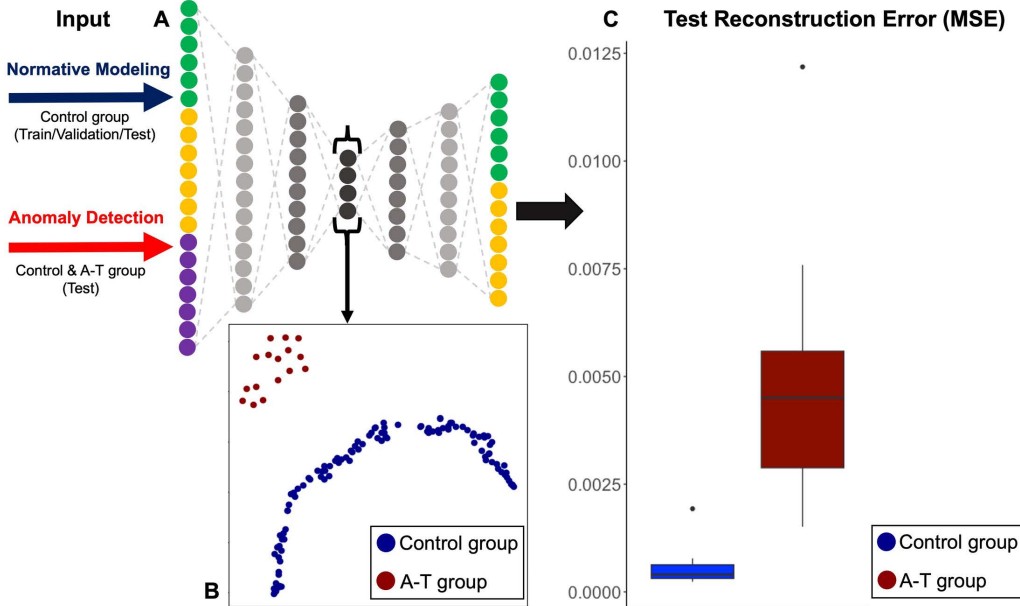

**Fig 2. (A) Schematic of the autoencoder architecture and overall research design. The autoencoder comprises seven asymmetrical, fully connected layers. The input layer includes 20 features: six perfusion features (green: caudate, hippocampus, pallidum, putamen, thalamus, cortical gray matter (CGM)), seven diffusion features (orange: hippocampus, pallidum, putamen, thalamus, CGM, cerebral white matter (CWM)), and seven covariates (purple: age, sex, cerebral volume, cerebellar volume, age², age × sex, age² × sex). The output layer contains 13 features, enabling reconstruction of only the six perfusion (green) and seven diffusion (orange) features.** *(B) UMAP embedding plot showing two dimensions for latent space encoding, colored by subject group (blue = control, red = A-T). (C) Boxplots for total reconstruction error difference in the final model for participants with A-T and controls, showing a significantly higher error for the A-T group (P-Wilcoxon = 1.5 × 10⁻⁶).*

the final model on the 16 A-T participants to detect anomalies based on the reconstruction errors associated with each feature, akin to Chamberland et al. [18]. The reconstruction errors obtained were compared across A-T participants and controls using a permutation-based t-test with 1,000 iterations.

To enhance model interpretability, we employed Shapley Additive Explanations (SHAP) [28], a method based on cooperative game theory that quantifies the contribution of each feature to the model predictions. SHAP isolates the effect of each feature (perfusion or diffusion input features) on each feature reconstruction (perfusion or diffusion output features) by considering the contribution of each feature across all possible combinations of features. The absolute SHAP value of an input variable indicates how much that variable influences the output, and its direction indicates whether an increase of the input feature increases (positive SHAP value) or decreases (negative SHAP values) the output. However, the SHAP value must the interpreted in the context of the distribution of the output variable. If the output is a probability value, then SHAP values between 0.5–1 can reflect a large impact, but if the distribution of the output variables ranges between 0 and 100, then a SHAP value of 5 can be deemed to be non-relevant. In our analysis, we excluded the self-effect of each feature, to isolate cross-feature influence. This approach enabled us to assess the importance of each input on each output individually and to interpret the reconstruction predictions of the autoencoder.

### 2.5. Association of reconstruction errors to clinical scores

We used reconstruction errors as predictors for clinical features according to the A-T NEST scores [19]. To assess these associations, we performed multivariate LASSO regression using reconstruction errors as independent variables, reporting normalized coefficients. All models were adjusted for age, age², sex, sex × age, sex × age², cerebral volume (mL), and cerebellar volume (mL), which were included as unpenalized covariates to preserve their a priori defined biological relevance [29]. We only reported coefficients with absolute magnitude > 0.1. As this is an exploratory study, multiple test correction was not applied [30].

## 3. Data sharing

The feature dataset of healthy control and A-T children used in this work, and the Python code to train and test the autoencoder have been made publicly available at https://github.com/stefanpsz/A-T_Autoencoder.

## 4. Results

### 4.1. Participants description

Sixteen A-T participants (median age [range] = 12.0 [4.6, 17.8] years, median cerebral volume = 1,030 mL, median cerebellar volume = 70 mL, F = 0.44) and 110 controls (22 from CATNAP and 88 from the Calgary dataset, median age [range] = 5.5 [2.9, 16.3] years, median cerebral volume = 1,085 mL, median cerebellar volume = 133 mL, F = 0.49) had usable data for both diffusion and perfusion sequences (**Fig 1**).

Of the 16 A-T and 22 control CATNAP participants, 15 and 21 participants, respectively, are part of a previously reported dataset [9]. This prior article reported classical statistical analyses using cerebellar diffusion data only whereas, in this manuscript, we report both diffusion and perfusion data analyzed with an autoencoder.

No group difference in sex (P = 0.90, $\chi^2$ Test) was found between A-T participants and controls. We found significant differences in age and cerebral and cerebellar volumes (P = $1.9 \times 10^{-5}$, $3.0 \times 10^{-2}$, $1.3 \times 10^{-14}$; T-test) between A-T participants and controls.

### 4.2. Session selection

For participants with more than one scanning session containing usable structural, diffusion, and perfusion data, the session in which the participant was the youngest was selected for CATNAP, whereas the one where the participant was the oldest was selected for the Calgary dataset. This selection minimized the difference in age range between both datasets.

### 4.3. Reconstruction errors of healthy controls and A-T participants

The autoencoder architecture and research design are illustrated in **Fig 2A**. Following training, we applied Uniform Manifold Approximation and Projection (UMAP) to reduce the dimensionality of the latent space from four to two dimensions, enabling visual inspection of clustering patterns. As shown in **Fig 2B**, UMAP embeddings revealed a distinct separation between controls and A-T participants, with each group forming well-defined clusters.

We then evaluated the reconstruction errors for individual features. Controls showed accurate reconstructions with minimal errors, whereas A-T participants exhibited significantly higher errors, reflecting deviations from the normative model. Overall, A-T participants had a median reconstruction error of $4.5 \times 10^{-3}$ (range: $1.5 \times 10^{-3}$, $1.2 \times 10^{-2}$), compared to controls ($4.0 \times 10^{-4}$, range: $2.4 \times 10^{-4}$, $1.9 \times 10^{-3}$), $P = 1.5 \times 10^{-6}$, Wilcoxon test (**Fig 2C**).

Detailed analysis of individual features revealed consistent trends across groups (**Fig 3**). Regarding perfusion reconstruction, A-T participants demonstrated significant overestimation of caudate (+5.8 mL/100 g/min, P < 0.001) and putamen (+1.3 mL/100 g/min, P = 0.013), together with significant underestimation of pallidum (−5.1 mL/100 g/min, P < 0.001), hippocampus (−1.3 mL/100 g/min, P = 0.019) and thalamus (−1.3 mL/100 g/min, P = 0.022). Interestingly, test controls showed a significant underestimation of hippocampus perfusion (−1.7 mL/100 g/min, P = 0.020).

Regarding diffusion reconstruction, A-T participants showed a significant underestimation of CGM ($-553 \times 10^{-6}$ mm²/s, P < 0.001) and CWM ($-175 \times 10^{-6}$ mm²/s, P < 0.001), and a significant overestimation of the hippocampus ($+73 \times 10^{-6}$ mm²/s, P < 0.001), caudate ($+63 \times 10^{-6}$ mm²/s, P = 0.013) and putamen ($+24 \times 10^{-6}$ mm²/s, P = 0.010).

### 4.4. Model explainability using shapley additive explanation (SHAP) values

Regarding model explainability, SHAP values highlighted consistent patterns in how each input feature influenced each output feature.

SHAP values for output perfusion features revealed that most output features, particularly caudate, putamen, hippocampus, CGM, and thalamus perfusion, were negatively affected by input perfusion and diffusion features (**Fig 4A**). Output perfusion features were positively affected by CGM diffusion and pallidum perfusion, and marginally by putamen and

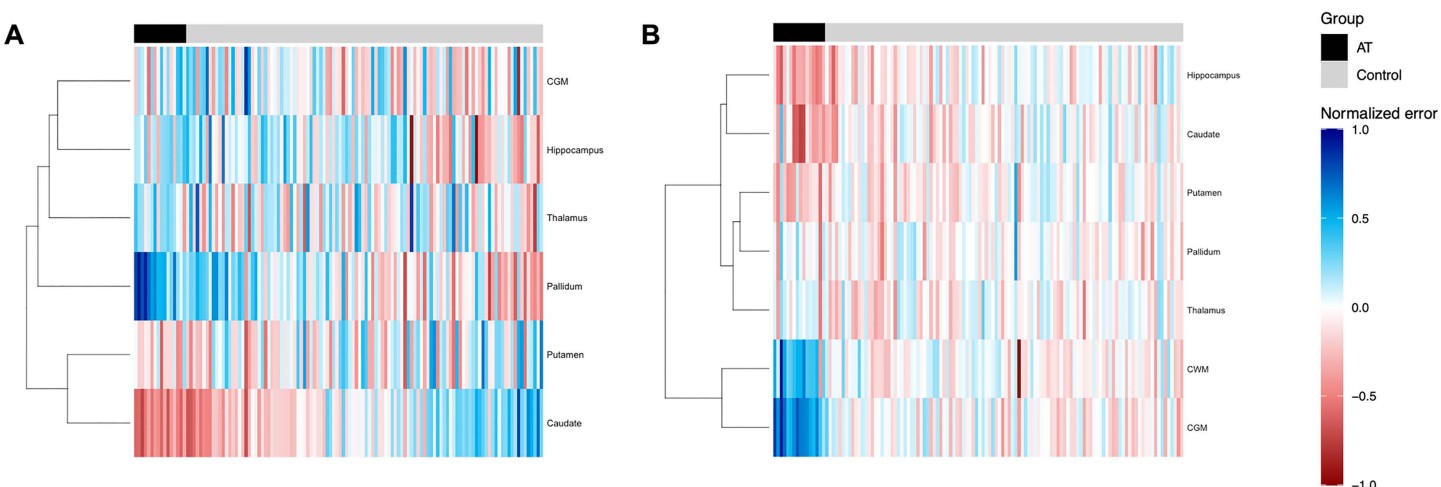

**Fig 3. Heatmap anomaly score of perfusion (A) and diffusion (B) reconstruction values for controls (grey) and A-T group (black).** Hierarchical clustering for feature clustering: Lance-Williams formula based on maximum linkage distance. Positive (blue) and negative (red) z-scores (using control reconstructed data as the reference group). Darker tones denote higher relative magnitude.

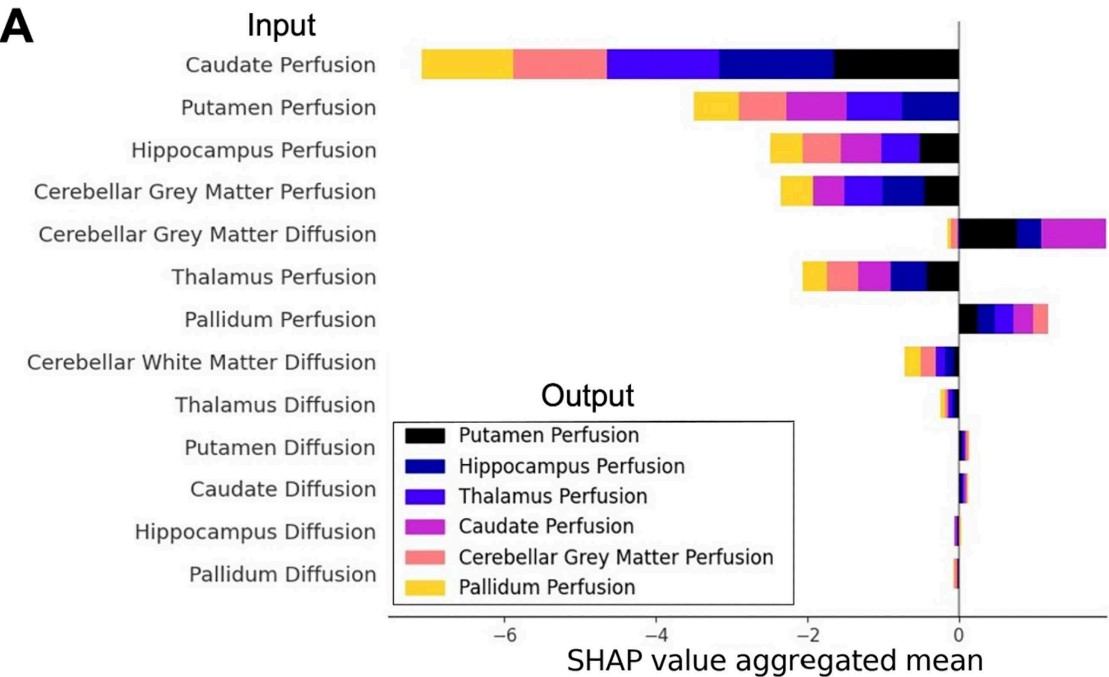

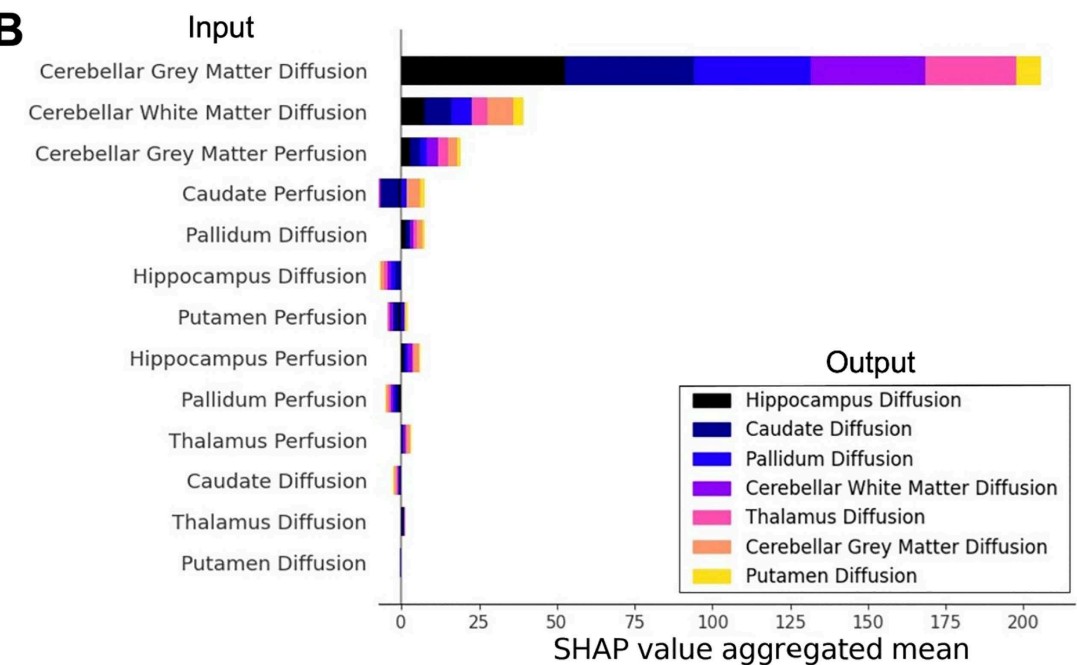

**Fig 4. Summary plots for SHAP values of perfusion (A) and diffusion (B) feature reconstruction errors.** Each input feature is represented by a horizontal bar, illustrating the total mean SHAP value attributed to every other feature. We employ color-coded sub-bars within each input bar to identify the mean SHAP values associated with different output variables. Positive means are stacked on the positive side of the x-axis. Negative means are stacked on the negative side of the x-axis.

caudate diffusion. The putamen, hippocampus, and thalamus showed the greatest variations attributed to the changes in input features.

SHAP values for output diffusion features revealed a mixture of positive and negative contributions from perfusion and diffusion input features (**Fig 4B**). Positive effects in diffusion features were mainly explained by CGM and CWM diffusion, as well as CGM perfusion. Low-magnitude positive effects were explained by pallidum diffusion and hippocampus perfusion. Negative effects were mainly explained by hippocampus diffusion, and putamen and pallidum perfusion. The hippocampus, caudate, and pallidum diffusion showed the greatest variations attributed to the value variations in input features.

For both perfusion and diffusion output features, the direction of the relationships shown for each of the individual input features (indicated by either positive or negative aggregated mean SHAP values) was largely consistent. However, several exceptions to this were noted. For example, for the CGM diffusion input, both the caudate and CGM perfusion outputs had negative mean SHAP values, whereas the other perfusion outputs had positive mean SHAP values. Similarly, both the caudate diffusion and putamen diffusion inputs showed positive and negative mean SHAP values for the relevant diffusion outputs. While speculative explanations could be proposed whereby disease-related loss of either excitatory or inhibitory actions of input structures differentially affects the outputs, in view of the small sample size we suspect these inconsistencies in the direction of the mean SHAP values are likely to be spurious.

### 4.5. Association of reconstruction errors to clinical scores

The association of reconstruction errors with clinical scores from the A-T NEST battery is shown in **Fig 5**. Ataxia was positively associated with increased reconstruction error in caudate perfusion ($\beta = 0.56$) and putamen perfusion ($\beta = 0.58$), and negatively associated with pallidum perfusion ($\beta = -0.50$), as well as with pallidum diffusion ($\beta = -0.37$), putamen diffusion ($\beta = -0.38$), and CGM diffusion ($\beta = -0.37$). Bradykinesia was positively associated with reconstruction error in caudate perfusion ($\beta = 0.75$) and thalamus perfusion ($\beta = 0.44$), and negatively associated with pallidum perfusion ($\beta = -1.63$), pallidum diffusion ($\beta = -0.89$), putamen diffusion ($\beta = -0.60$), and CWM diffusion ($\beta = -0.23$).

Hypermotor scores showed negative associations with reconstruction error in hippocampus perfusion ($\beta = -0.55$), CGM perfusion ($\beta = -0.55$), caudate diffusion ($\beta = -0.73$), putamen diffusion ($\beta = -0.77$), thalamus diffusion ($\beta = -0.77$), CGM diffusion ($\beta = -1.10$), and CWM diffusion ($\beta = -0.55$), while positive associations were observed with caudate perfusion ($\beta = 0.26$) and putamen perfusion ($\beta = 0.52$). Dystonia was associated with caudate perfusion ($\beta = 0.29$), hippocampus perfusion ($\beta = -0.55$), and pallidum perfusion ($\beta = -0.44$) reconstruction errors.

## 5. Discussion

In this exploratory study, we characterized subcortical anomalies in children with A-T using an anomaly detection approach based on a deep self-supervised normative model of healthy controls. We compared the difference in reconstruction errors of subcortical and cerebellar perfusion and diffusion data of A-T participants and controls. We used an explainability framework to understand which features drive anomalies in participants with A-T and performed a statistical analysis to show significant correlations of feature reconstruction errors with hypermotor symptoms, bradykinesia, dystonia, and ataxia.

Our analysis showed a clear separation of healthy controls from participants with A-T. This separation was evident in the autoencoder latent space (**Fig 2B**). Data was correctly reconstructed in controls but not in participants with A-T, who generally showed significantly higher errors. This was especially true for pallidum and caudate perfusion, as well as for CWM and CGM diffusion (**Fig 3**). Overall, our results are indicative of the over- and under-estimations that separate A-T participants and controls, especially in the aforementioned ROIs.

Cerebellar atrophy is an established finding in A-T, with previous quantitative imaging studies showing reduced cerebellar volume, increased ADC, and altered metabolites measured by magnetic resonance spectroscopy [9,31–33]. Therefore, our analysis was expected to identify altered CWM and CGM diffusion, and the clear separation between children with A-T

and controls provides reassurance that the reconstruction error-based analysis approach performs well in differentiating participants with and without the disease.

The identified differences in reconstruction error for MRI-based perfusion measures from basal ganglia structures between children with and without A-T are novel findings. Previous research has provided limited evidence indicating basal ganglia abnormalities in people with A-T. A PET study by Volkow and colleagues demonstrated a 16% increase ($P < 0.05$) in the globus pallidus of young adults with A-T compared to age-matched healthy controls [12]. A strong correlation ($r = 0.74$) between globus pallidus FDG uptake and motor scores was observed, although their motor assessment did not isolate extrapyramidal components. Koepp and colleagues reported a child with A-T with severe progressive dystonia

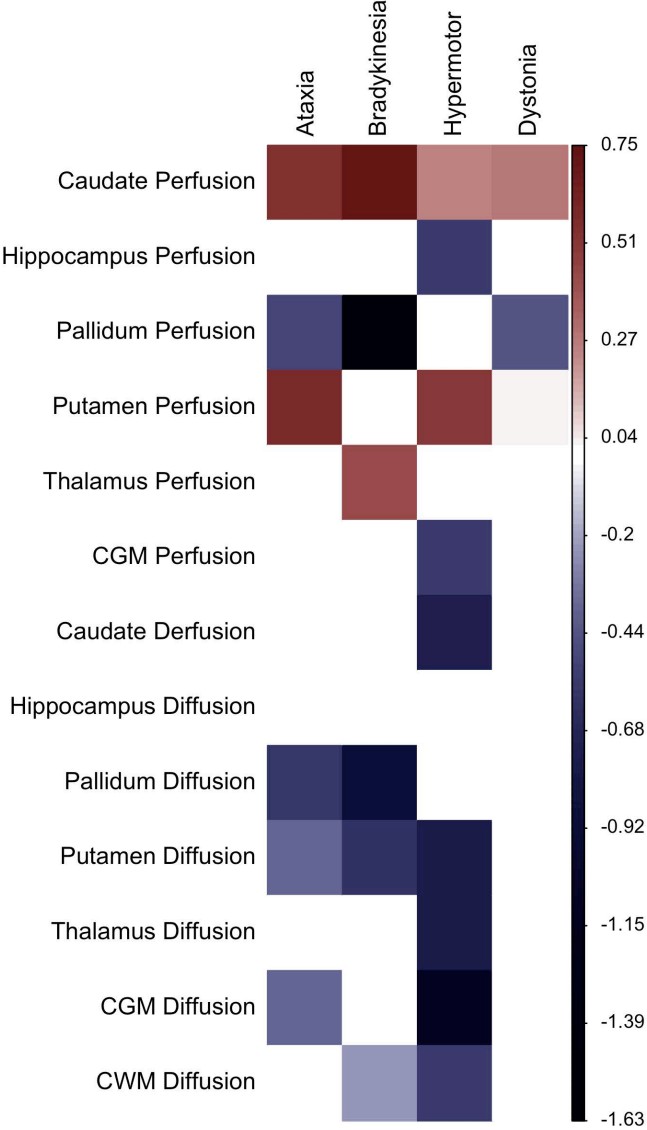

**Fig 5. Clinical correlations heatmap.** Associations of the brain imaging features' reconstruction errors with clinical scores adjusted by age, sex, and cerebral volume. Colors denote the standardized regression coefficient directions (red = positive, blue = inverse) and magnitudes (darker colors = stronger magnitude). * P<0.05, ** P<0.01.

showing reduced D2 receptor binding in the striatum on PET imaging [34], and a case report describes improvements in dystonia following deep brain stimulation insertion targeting the globus pallidus interna of an adult with A-T [35].

Similarly to this work, recent developments of AI in movement disorders have shown the potential of these tools to characterize and diagnose these diseases [36]. For example, AI has been used for prediction and diagnosis in different types of ataxias [37–41], for diagnosis of essential tremor [42], and to model progression of Huntington's Disease [43]. Moreover, "black box" AI models can be made more transparent by using explainability techniques such as SHAP. In our work, SHAP analysis revealed consistent trends among perfusion and diffusion features explanatory influence. In the case of perfusion, the most influential explanatory features were those associated with the putamen, hippocampus, and thalamus (**Fig 4A**). In the case of diffusion, the main explanatory features were those associated with the hippocampus, caudate, and pallidum (**Fig 4B**).

In A-T patients, SHAP analysis reveals how input features drive anomalies in reconstructed outputs, with contributions interpreted relative to the scales of perfusion (0–100 mL/100 g/min) and diffusion (0–3,000 × 10$^{-6}$ mm²/s) and reconstruction errors (1–5 mL/100 g/min for perfusion, tens to hundreds ×10$^{-6}$ mm²/s for diffusion). For example, a negative SHAP value for caudate on putamen perfusion indicates that elevated caudate perfusion in A-T corresponds to a reduced putamen perfusion estimate compared to controls, suggesting disrupted inter-regional dynamics. This feature-specific insight refines our understanding of the anomalies observed, complementing broader network-level findings.

The SHAP analysis showing that perfusion or diffusion values of cerebellar cortical or supratentorial sub-cortical gray matter structures can be 'explained' to differing extents by the perfusion or diffusion values in others of these structures suggests a network-level effect of A-T. Indeed, as well as the well-established classical motor control circuits involving the basal ganglia and thalamus, recent evidence also demonstrates the presence of direct cerebellar-basal ganglia and cerebellar-thalamic circuits [44]. These provide a potential biological substrate for disease-related network-level alterations in diffusion (reflecting tissue ultrastructure) and metabolically coupled perfusion parameters.

However, it is important to recognize the fundamental limitation of the SHAP explainability analysis which is the uncertainty in determining the direction of the observed correlations. While associations are evident, this analysis does not reveal causality. Thus, it is unclear which features cause changes in others, raising the possibility that the relationships observed may be due to confounding variables rather than actual causal effects. For example, we note that the hippocampus, included in the analysis to provide a non-motor supratentorial subcortical structure, was found to be an explanatory feature for diffusion and perfusion alterations in other subcortical structures participating in motor control circuits. This was an unexpected and possible spurious finding or could potentially relate to the recently described interplay between the cerebellum and hippocampus [45]. The limitation of the SHAP explainability analysis emphasizes the need for caution in interpreting the results. Future research could employ methods such as functional analysis, time series analysis, and longitudinal studies to better understand the underlying causal mechanisms.

Another limitation of the present study is that although state of consciousness (awake vs sleep) can have an impact on brain perfusion [46], we did not take this into account because none of the A-T participants and only 6 of the 110 controls subjects that were considered were sleeping during scanning.

We explored the relationships between the diffusion and perfusion imaging features and neurological scores for ataxia, bradykinesia, hypermotor, and dystonia. We identified several correlations involving the basal ganglia and cerebellar ROIs. Notably, we found negative correlations of hypermotor with CGM diffusion reconstruction error and of dystonia with pallidum perfusion reconstruction error. The results from this analysis should be considered as preliminary; as this is an exploratory study, we have not applied multiple test correction and the sample size is small, hence the observed associations may be spurious and are not necessarily directly causally related. Despite the individualized nature of the analyses that our model provides for A-T subjects, the conclusions might not be completely generalizable due to the modest number of A-T participants used. Larger studies are required to confirm the observed associations. In addition to the small size of the A-T cohort, a limitation of this work is the relatively small sample size of the normative dataset for training.

To our knowledge, this study is the first to find imaging correlates of movement disorders in A-T linked to MRI-based perfusion and diffusion metrics. These findings should prompt further investigation of the nature of basal ganglia pathology in A-T as structural or network components that could be used as future therapeutic targets. More broadly, we demonstrate that normative autoencoder models are adept at handling complex, varied data and could be particularly beneficial in tailoring individualized interventions. Our study also introduces a novel approach by utilizing tabular data within an autoencoder framework, a method not commonly explored in prior research.

The development of autoencoder-based normative models based on healthy controls shows promise for advancing healthcare, especially in scenarios where the availability of patient data is limited. Future work should strengthen these findings by recruiting a larger cohort of controls and participants with A-T, emphasizing individuals of similar ages to enhance the homogeneity of the study group. Future work should also employ directional explainability analysis to understand the causal relationships between features.

## Supporting information

**S1 Fig. Median Mean Squared Error (MSE) loss comparison among the five best model sets for training (80 controls), validation (20 controls), test (10 controls), and A-T set (16 participants with A-T).** The median MSE illustrates the loss evolution for each group for the five best models and its interquartile range (IQR).
(TIF)

**S1 File. Appendix A1 and Appendix A2.**
(DOCX)

**S2 File. Inclusivity-in-global-research-questionnaire.** CATNAP1 research ethics committee approval letter. CATNAP2 research ethics committee approval letter.
(DOCX)

## Acknowledgments

The authors thank the children, young people, and families participating in the CATNAP study. We gratefully acknowledge the support of Dr. Gabby Chow, Dr. Manish Prasad, Dr. Min Ong, and Dr. Jeyanthi Rangaraj for support with neurological assessments; Hannah McGlashan and Dr. Felix Raschke for assistance with recruiting participants and Andrew Cooper for assistance with performing the MRI scans.

## Author contributions

**Conceptualization:** Stefan Pszczolkowski.

**Data curation:** Caroline C.V. Blanchard, William P. Whitehouse, Rob A. Dineen.

**Formal analysis:** Catalina Saini, Cristian Salazar-Vilches.

**Funding acquisition:** William P. Whitehouse, Rob A. Dineen.

**Investigation:** Stefan Pszczolkowski.

**Methodology:** Denis Parra, Stefan Pszczolkowski.

**Resources:** William P. Whitehouse.

**Software:** Catalina Saini, Cristian Salazar-Vilches.

**Supervision:** Denis Parra, Rob A. Dineen, Stefan Pszczolkowski.

**Validation:** Catalina Saini, Cristian Salazar-Vilches.

**Writing – original draft:** Catalina Saini, Cristian Salazar-Vilches, Denis Parra, Rob A. Dineen, Stefan Pszczolkowski.

**Writing – review & editing:** Catalina Saini, Cristian Salazar-Vilches, Caroline C.V. Blanchard, William P. Whitehouse, Denis Parra, Rob A. Dineen, Stefan Pszczolkowski.

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
