## [Decision Letter · Decision Letter 0]

9 Apr 2025

PONE-D-25-03845Personalized MRI-based Characterization of Subcortical Anomalies in Ataxia-Telangiectasia Using Deep-LearningPLOS ONE

Dear Dr. Pszczolkowski Parraguez,

Thank you for submitting your manuscript to PLOS ONE. After careful consideration, we feel that it has merit but does not fully meet PLOS ONE’s publication criteria as it currently stands. Therefore, we invite you to submit a revised version of the manuscript that addresses the points raised during the review process.

We look forward to receiving your revised manuscript.

Kind regards,

Alpen Ortug, PhD

Academic Editor

PLOS ONE

Journal Requirements:

Reviewers' comments:

Reviewer's Responses to Questions

**Comments to the Author**

1. Is the manuscript technically sound, and do the data support the conclusions?

Reviewer #1: Yes

Reviewer #2: Yes

2. Has the statistical analysis been performed appropriately and rigorously? 

Reviewer #1: Yes

Reviewer #2: Yes

3. Have the authors made all data underlying the findings in their manuscript fully available?

Reviewer #1: Yes

Reviewer #2: Yes

4. Is the manuscript presented in an intelligible fashion and written in standard English?

Reviewer #1: Yes

Reviewer #2: Yes

5. Review Comments to the Author

Reviewer #1: The authors utilize a novel method, a normative self-supervised deep autoencoder, which they trained on MRI diffusion and perfusion sequences from healthy controls. They then apply this trained model to a set of patients with ataxia-telangiectasia and identify regions where reconstruction errors occurred, highlighting regions of brain pathology in this disease. Before reading this submission, I was not aware of this methodology, and the writing is clear and the reasoning well explained in the manuscript. The findings are well described, as are the figures. Moreover, the discussion is clear and highlights the limitations, namely that the directionality of the findings is unknown. Moreover, no pathophysiologic target can be identified using these methods, only a brain region that is affected. Overall, I have no editorial suggestions and think this paper is worthy of publication with little or no additional editing.

Reviewer #2: The study presents an investigation into the basal ganglia (motor circuits) in Ataxia-Telangiectasia (A-T) patients, utilizing a normative self-supervised deep autoencoder trained to highlight deviations in imaging features (in this case perfusion and diffusion) compared to healthy controls. The results are positioned as exploratory and may serve as a foundation for future in-depth investigations. Below are specific comments regarding different sections of the manuscript.

Introduction

- The motivation for investigating perfusion and diffusion differences in A-T patients could be better articulated. Are there specific biological mechanisms or prior findings that suggest these parameters would be affected? Providing a clearer rationale would strengthen the introduction.

Methods

- The manuscript states that only imaging features are reconstructed using the model. Additional details on how covariates (e.g., age, sex) are incorporated or accounted for within the model would help clarify their impact on model performance.

- It is unclear whether the state of participants during scanning (awake vs. asleep) was documented. Brain perfusion can vary significantly based on the state of consciousness (and therefore potentially between controls and patients), and this factor should ideally be addressed in the methods section.

Results/figures

- Further information is needed on how age, sex, and cerebral volume influenced the correlation between reconstruction errors (this is related to my earlier point on the inclusion of covariates in the model) and clinical scores. These factors could have significant effects on the findings and should be reported.

- The interpretation of differences between controls and A-T patients based on the UMAP representations should be elaborated upon. What insights could these representations provide regarding differences between patients and controls in terms of features?

- Figure 3 would benefit from a clear color legend or bar to enhance interpretability.

- The x-axis label of figure 4 is confusing. As these are stacked bar plots, should this not be the sum of the means instead of (just) mean?

Discussion

- The use of SHAP analysis is an interesting approach to better understand the model output; however, it remains somewhat unclear how results from different structures could influence each other. Additional explanation on how these changes relate to the brain’s subcortical motor circuits would be helpful.

- The SHAP analyses further suggests that input features contributed equally (positively or negatively) in most cases to the output features, except for cerebellar gray matter diffusion (for perfusion output) and caudate and putamen perfusion (for diffusion output). This is also reported in section 4.4 of the results, but is not elaborated further on in the discussion. This observation warrants further discussion.

- Finally, the differences between positive and negative SHAP values should be explained further. Can these differences be interpreted as their impact to be increasing or decreasing the perfusion/diffusion estimates, respectively? A clearer interpretation would strengthen the discussion.

6. PLOS authors have the option to publish the peer review history of their article (what does this mean? ). If published, this will include your full peer review and any attached files.

**Do you want your identity to be public for this peer review?** For information about this choice, including consent withdrawal, please see our Privacy Policy .

Reviewer #1: No

Reviewer #2: **Yes: ** Roy Haast

---

## [Author Response · Author response to Decision Letter 1]

13 Jun 2025

We would like to thank the two reviewers for their kind and insightful comments. We are sure that by addressing them we have strengthen and improved our manuscript substantially.

We have included in this submission a "Response to Reviewers" document where we address each of the questions/comments raised by them.

---

## [Decision Letter · Decision Letter 1]

8 Jul 2025

Personalized MRI-based Characterization of Subcortical Anomalies in Ataxia-Telangiectasia Using Deep-Learning

PONE-D-25-03845R1

Dear Dr. Pszczolkowski Parraguez,

We’re pleased to inform you that your manuscript has been judged scientifically suitable for publication and will be formally accepted for publication once it meets all outstanding technical requirements.

Kind regards,

Alpen Ortug, PhD

Academic Editor

PLOS ONE

Additional Editor Comments (optional):

Reviewers' comments:

Reviewer's Responses to Questions

**Comments to the Author**

1. If the authors have adequately addressed your comments raised in a previous round of review and you feel that this manuscript is now acceptable for publication, you may indicate that here to bypass the “Comments to the Author” section, enter your conflict of interest statement in the “Confidential to Editor” section, and submit your "Accept" recommendation.

Reviewer #2: All comments have been addressed

2. Is the manuscript technically sound, and do the data support the conclusions?

Reviewer #2: Yes

3. Has the statistical analysis been performed appropriately and rigorously? 

Reviewer #2: Yes

4. Have the authors made all data underlying the findings in their manuscript fully available?

Reviewer #2: No

5. Is the manuscript presented in an intelligible fashion and written in standard English?

Reviewer #2: Yes

6. Review Comments to the Author

Reviewer #2: The authors have addressed all the previous concerns that I have. Therefore I support publication in PLOS One.

7. PLOS authors have the option to publish the peer review history of their article (what does this mean? ). If published, this will include your full peer review and any attached files.

**Do you want your identity to be public for this peer review?** For information about this choice, including consent withdrawal, please see our Privacy Policy .

Reviewer #2: **Yes: ** Roy AM Haast

---

## [Editor Report · Acceptance letter]

PONE-D-25-03845R1

PLOS ONE

Dear Dr. Pszczolkowski,

I'm pleased to inform you that your manuscript has been deemed suitable for publication in PLOS ONE. Congratulations! Your manuscript is now being handed over to our production team.

Kind regards,

on behalf of

Dr. Alpen Ortug

Academic Editor

PLOS ONE